# Model Predictive Current Control of an Induction Motor Considering Iron Core Losses and Saturation

Mateo Bašić *[ID], Dinko Vukadinović [ID] and Ivan Grgić [ID]

Department of Power Engineering, Faculty of Electrical Engineering, Mechanical Engineering and Naval Architecture, University of Split, 21000 Split, Croatia; dvukad@fesb.hr (D.V.); igrgic00@fesb.hr (I.G.)
* Correspondence: mabasic@fesb.hr

**Abstract:** The paper considers the model predictive current control (MPCC) of an induction motor (IM) drive and evaluates five IM models of different complexities—from conventional to magnetic saturation, iron losses, and stray-load losses—for the MPCC design. The validity of each considered IM model and the corresponding MPCC algorithm is evaluated by comparison of the following performance metrics: the total harmonic distortion of the stator current, the average switching frequency, the rotor flux magnitude error, the rotor flux angle error, and the product of the first two metrics. The metrics' values are determined in wide ranges of the rotor speed (0.1–1 p.u.) and load torque (0–1 p.u.) through simulations performed in the MATLAB Simulink environment. The obtained results allow us to identify the IM model that offers the best tradeoff between the practicability and accuracy. Furthermore, a control effort penalization (CEP) is suggested to reduce the average switching frequency and, hence, the power converter losses. This involves constraining the simultaneous switching to a maximum of two branches of the three-phase power converter, as well as inclusion of the weighted switching penalization term in the cost function. Finally, the performance—both steady-state and dynamic—of the proposed MPCC system with CEP is compared with that of the analogous field-oriented controlled (FOC) IM drive. The inverter switching frequency is reduced more than twice by including the frequency-dependent iron-loss resistance in the MPCC. It is additionally reduced by implementing the proposed CEP strategy without sacrificing many other performance metrics, thus achieving a performance comparable to the FOC IM drive.

**Keywords:** model predictive control; induction motor; iron losses; stray-load losses; magnetic saturation; dynamic model; field-oriented control; voltage source inverter

## 1. Introduction

With the recent developments in electromobility, the control of motor drives has gained increased relevance and importance despite being one of the most traditional electrical engineering topics. In high-performance induction motor (IM) drives, two control strategies are dominating the market, namely field-oriented control (FOC) [1] and direct torque control (DTC) [2]. FOC ensures decoupled control of the IM torque and flux, as in the direct current (DC) machine, and typically implies a cascaded structure with the use of proportional-integral (PI) controllers and pulse-width modulation (PWM), leading to a good steady-state and dynamic performance. DTC is characterized by a simpler structure since it does not require current control nor the PWM stage. Instead, it directly imposes the most appropriate voltage vector selected from a predefined look-up table, resulting in a faster dynamic response. On the downside, a poor low-speed operation, variable switching frequency (due to the application of hysteresis controllers), and high torque ripples are associated with this type of control.

Until the early 2000s, model predictive control (MPC) systems—a sophisticated control technique that was first established in the process industry in the 1970s [3]—did not receive much attention in power electronics. This was mostly due to a lack of powerful

computing resources. However, the availability of microprocessors, digital signal processors (DSPs), and field-programmable gate arrays (FPGAs) with enhanced computational capabilities at reduced costs has reignited the power electronics community's interest in MPC technologies, ranging from low- to high-power applications [4–7] and resulting in an exponential growth in the number of annual publications [8]. The appeal of the MPC lies in its intuitiveness, simple implementation for nonlinear problems, flexibility, and high performance. Particularly, the *direct* MPC (DMPC) (also known as the *finite control set* MPC) has been favored in the power electronics community due to its design simplicity and the possibility to include system constraints [5]. The *indirect* MPC (also known as the *continuous* MPC), on the other hand, offers constant switching frequencies and is gaining attention, especially for grid-tie converters, but it requires a PWM modulator [9,10]. In DMPC, the future behavior of the system is predicted based on the corresponding model and a finite number of switch positions, whereas the optimal position is found through minimization of the cost function and *directly* applied to the converter, thus eliminating an intermediate PWM stage. Two major subtypes of DMPC can be identified in IM drive applications, namely model predictive torque control (MPTC) [11,12] and model predictive current control (MPCC) [13]. The experimental comparison of these methods was carried in [14,15], where it was concluded that the MPCC method has lower computational time, fewer tuned parameters, lower current distortion, and higher robustness to stator resistance variations. The MPTC method, on the other hand, has lower torque ripples and higher robustness to magnetizing inductance variations. Other than that, both methods provide very good and similar performances.

MPC techniques, by definition, rely on the model of the system under control. Therefore, it is susceptible to mismatches between the system behavior predicted by its model and the behavior of the actual system. This involves both discrepancies in the system parameters' values as well as in the accounted phenomena. There are many different IM model variants in the literature but the most common is the conventional IM model [16], which is also most often used for the MPCC. The conventional IM model assumes, among other things, a magnetic circuit that is linear, electrically non-conductive, and lossless. Hence, it does not account for the magnetic saturation, the iron losses, or the stray-load losses (SLLs). However, efforts have been made to include some of these phenomena in MPC-related studies.

In [17], a loss minimization strategy based on the IM model with included iron losses and magnetic saturation was considered as part of the MPTC-based system, in which a combination of the DMPC and the dead-beat method was used. The iron-loss resistance was represented as either a constant or frequency-dependent parameter, whereas the magnetizing inductance was modeled as a function of the magnetizing current. Still, these phenomena were not considered in the MPTC algorithm, but only in the loss-minimization algorithm. In [18], a model predictive control of α-β components of the stator flux vector was proposed. Again, iron losses were considered for loss minimization through rotor flux reference adjustment, but they were neglected in the MPC algorithm, whereas magnetic saturation was not at all considered. A variant of the MPTC system with three PI controllers was considered in [19] for speed sensorless IM control. Both the speed observer and loss minimization algorithm included a frequency-dependent iron-loss resistance. Magnetic saturation was, however, neglected, which may be considered justified given the fact that MPTC is not much sensitive to magnetizing inductance variations. Another MPTC-based system including iron losses was considered in [20]. Although the loss minimization algorithm accounted for the iron losses, they were neglected in the MPC algorithm, as well as magnetic saturation. In [21], both magnetic saturation and iron losses were considered. A time varying iron-loss resistance was modeled as a function of the magnetizing flux and its time derivative. A cost function was used by which an optimal pair of rotor flux vectors is sought and applied using the corresponding reference voltage vectors in the given switching period—each in its corresponding share. However, this was implemented in the context of *indirect* MPC. An MPC variant in which the cost function contains the

absolute differences of the rotor speed and flux was proposed in [22]. An optimized genetic algorithm (GA) was used for online estimation of IM parameters, including the iron-loss resistance and magnetizing inductance, utilized by MPC. In [23], loss minimization based on the IM model was considered, but both the iron losses and magnetic saturation were neglected in the MPC algorithm. This was the study preceding [20], so similar considerations apply. Lastly, in [24], the model predictive power control (MPPC) was considered for a doubly fed induction generator, combined with a model-based loss minimization. However, iron losses were neglected in the MPC algorithm and magnetic saturation was not even considered in the paper.

In all of the above-mentioned studies that consider iron losses, the corresponding resistance is placed in parallel to the magnetizing inductance within the IM equivalent circuit. This increases the number of differential equations used to describe the IM behavior and, thus, implies a higher computational cost. In contrast, by placing the iron-loss resistance in parallel to the stator inductance, as proposed in [25], the number of differential equations stays the same as in the conventional IM model. The same goal is achieved by conveniently placing the SLL resistance in series with the stator resistance, as proposed in [26], instead of placing it in parallel with the leakage inductances. Still, neither of these two studies considered magnetic saturation. In addition, the SLLs are neglected in [25], whereas in [26], the iron-loss resistance is placed in parallel to the magnetizing inductance. In [27], a dynamic IM model was proposed that combines the favorable features of the models from [25,26], and additionally includes magnetic saturation and variable stray-load and iron losses. Its validity was experimentally verified by using four IMs of different efficiency classes and rotor cage material. Due to its accuracy, compactness, and simplicity, this model was later used in [28–31] for the control detuning analysis, estimation of the winding resistances and rotor speed, and loss minimization of field-oriented and sliding-mode controlled induction machines.

In this study, the IM model proposed in [27] is utilized for the MPC design and performance analysis of the considered MPCC-based system. Some of the contributions are derived directly from this as follows:

- The proposed control algorithm is the first algorithm from the MPC group that is based on the IM model from [27]. This simplifies the corresponding equations greatly compared to similar advanced IM models while not sacrificing the accuracy too much. In general, a simpler model leads to a simpler control algorithm and a less expensive implementation.
- The proposed control algorithm is the first MPC algorithm that allows for inclusion of the IM magnetic saturation, iron losses, and SLLs. A more accurate model leads to a more accurate prediction of controlled variables and, hence, better control.
- The proposed control algorithm is the first algorithm from the *MPCC group* that includes *any* of the mentioned IM phenomena.
- The proposed control algorithm allows us to partially include the mentioned IM phenomena so different algorithms can be applied for different applications. The transition between the algorithms is straightforward and could be implemented online if required.

The objectives set in this study can be identified as follows:

- The level of the IM model's complexity that ensures both a practical and sufficiently accurate control algorithm is to be determined. This requires evaluating the necessity of accounting the mentioned phenomena as well as the way they should be accounted, all based on the selected performance metrices.
- After the IM model suitable for the selected application is identified, the impact of control effort penalization (CEP) on the performance metrics is to be evaluated.
- The steady-state and dynamic performance of the final proposed MPCC algorithm is to be evaluated against the industry-standard FOC method and the existing competing MPC algorithms.

The paper is structured as follows: Section 2 introduces the utilized dynamic IM model including the magnetic saturation, iron losses, and SLLs. In Section 3, an overview of the control system under consideration is given, including the MPCC algorithm and the FOC counterpart. The proposed model predictive controller is presented and elaborated in detail in Section 4, and simulation results are presented and discussed in Section 5. Finally, Section 6 concludes the paper.

## 2. Induction Machine Modeling

The space-vector equivalent circuit of the considered dynamic IM model is shown in Figure 1a, where the magnetic saturation is included through variable magnetizing inductance $L_m$, whereas the SLLs and the iron losses are included through variable resistances $R_{sll}$ and $R_m$, respectively. Due to the suggested convenient placement of these resistances within the equivalent circuit, the differential order of the simpler, conventional IM model is retained, thus restraining the inevitable rise in computational cost, e.g., as opposed to the configuration in which $R_m$ is placed in parallel with $L_m$ [26,32,33] or that in which $R_{sll}$ is placed in parallel with the leakage inductances $L_{sl}$ and/or $L_{rl}$ [32,34,35]. In this way, a balance is struck between the model's practicability and accuracy.

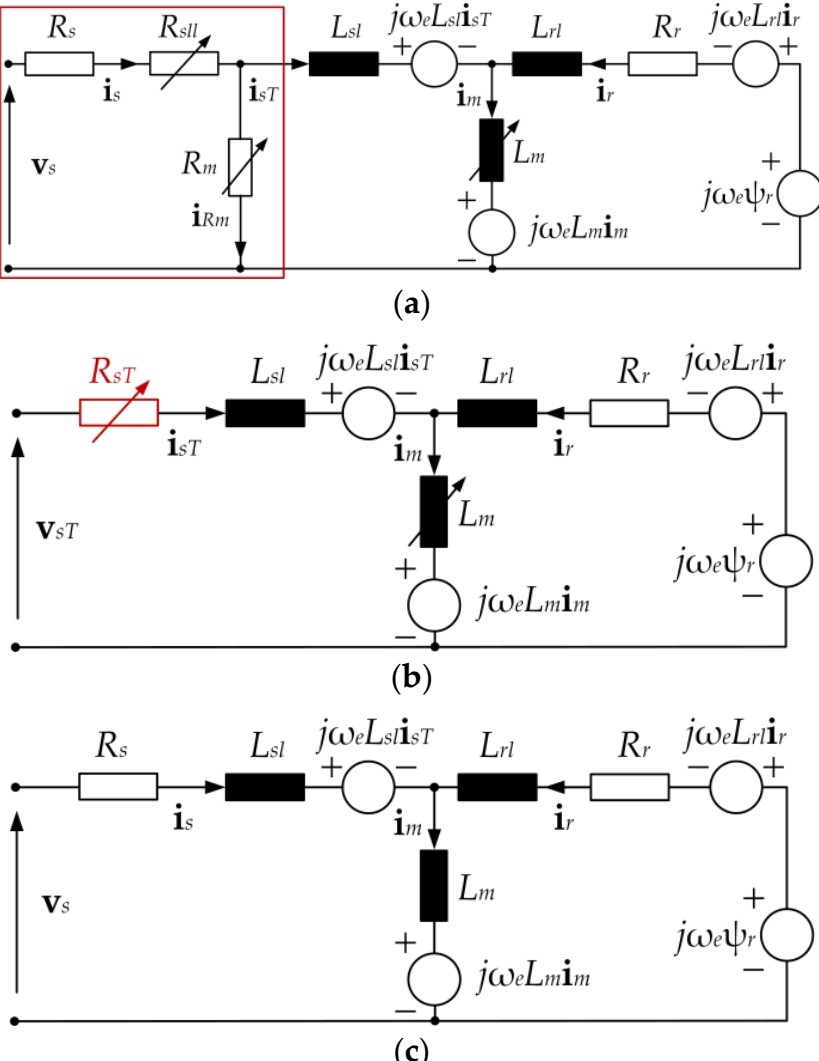

**Figure 1.** Space-vector equivalent circuits of dynamic IM models: (**a**) advanced model including magnetic saturation and variable stray-load and iron losses; (**b**) simplified representation of the advanced model; (**c**) conventional model.

Through the application of the Thevenin transform to the area enclosed by the red rectangle in Figure 1a, the resulting equivalent circuit (Figure 1b) takes a compact form of the well-known conventional IM model (Figure 1c).

The space-vector equations of the conventional IM model in Figure 1c in the synchronously rotating reference frame ($\omega_e$) are given as follows:

$$\mathbf{v}_s = R_s \mathbf{i}_s + \frac{d\boldsymbol{\psi}_s}{dt} + j\omega_e \boldsymbol{\psi}_s \tag{1}$$

$$0 = R_r \mathbf{i}_r + \frac{d\boldsymbol{\psi}_r}{dt} + j(\omega_e - \omega_r)\boldsymbol{\psi}_r \tag{2}$$

$$\boldsymbol{\psi}_s = (L_m + L_{sl})\mathbf{i}_s + L_m \mathbf{i}_r = L_s \mathbf{i}_s + L_m \mathbf{i}_r \tag{3}$$

$$\boldsymbol{\psi}_r = (L_m + L_{rl})\mathbf{i}_r + L_m \mathbf{i}_s = L_r \mathbf{i}_r + L_m \mathbf{i}_s \tag{4}$$

$$T_e = \frac{3}{2} p \frac{L_m}{L_r} \mathrm{Im}\left\{ \mathbf{i}_s \overline{\boldsymbol{\psi}_r} \right\} = T + J\frac{d\omega_r}{dt} + B\omega_r \tag{5}$$

where $\mathbf{v}_s$ is the stator phase voltage space vector; $R_s$ and $R_r$ are the stator and rotor resistances, respectively; $\mathbf{i}_s$ and $\mathbf{i}_r$ are the stator and rotor current space vectors, respectively; $\boldsymbol{\psi}_s$ and $\boldsymbol{\psi}_r$ are the stator and rotor flux linkage space vectors, respectively; $\omega_e$ and $\omega_r$ are the stator angular frequency and rotor angular speed, respectively; $L_m$, $L_s$, and $L_r$ are the magnetizing, stator, and rotor inductances, respectively; $T_e$ and $T$ are the electromagnetic and load torque, respectively; $p$ is the number of pole pairs; the line above the symbol denotes complex conjugation; $J$ is the moment of inertia; and $B$ is the viscous friction coefficient ($B = 0$ in this study).

Note that (1)–(5) are also valid for the IM model presented in Figure 1a,b, provided that $R_s$, $\mathbf{v}_s$, and $\mathbf{i}_s$ are, respectively, substituted by

$$R_{sT} = \frac{R_m(R_s + R_{sll})}{R_s + R_{sll} + R_m} \tag{6}$$

$$\mathbf{v}_{sT} = \mathbf{v}_s \frac{R_m}{R_s + R_{sll} + R_m} \tag{7}$$

$$\mathbf{i}_{sT} = \mathbf{i}_s \frac{R_s + R_{sll} + R_m}{R_m} - \frac{\mathbf{v}_s}{R_m} \tag{8}$$

The space-vector equations of the IM model shown in Figure 1a are provided in Appendix A. It is observed in (6)–(8) that by setting $R_{sll} = 0$ and $R_m \to$ inf., it follows $R_{sT} = R_s$, $\mathbf{v}_{sT} = \mathbf{v}_s$, and $\mathbf{i}_{sT} = \mathbf{i}_s$.

### 2.1. Iron Loss Modeling

IM's iron losses are commonly separated into hysteresis and eddy-current losses, with the latter being typically dominant at frequencies of the order of a few kHz [36,37]. Since model predictive control of IMs is, like field-oriented control, based on the corresponding fundamental space-vector equations, with the fundamental frequency typically being of the order of a few tens of Hz, eddy-current losses are neglected in this study to simplify the analysis. Additionally, since $R_{sll}$ is, in Figure 1a, connected in series with $R_s$, it contributes to the no-load losses, so the $R_m$ value determined from the standard no-load test needs to be adjusted accordingly when this model is utilized, as explained in [27]. Evidently, this is not the case if the SLLs are neglected (i.e., $R_{sll} = 0$). In both cases, the iron-loss resistance can be represented as a function of the stator angular frequency and flux magnitude as follows [27]:

$$R_m(\omega_e, \psi_s) = R_{m-rated} \frac{6\pi^2}{K_h(\psi_s)} \cdot \frac{\omega_e}{\omega_{e-rated}} \tag{9}$$

where $R_{m-rated}$ is the $R_m$ value at rated frequency and magnetization (for the considered IM, $R_{m-rated}$ = 1258.3 Ω for accounted SLLs, and $R_{m-rated}$ = 1012.3 Ω for neglected SLLs), $\omega_{e-rated}$ denotes the rated stator angular frequency, and $K_h(\psi_s)$ is the stator flux-dependent hysteresis loss coefficient (given in Appendix B), whose value differs depending on whether the SLLs are accounted or not [27].

Figure 2 shows the iron-loss resistance characteristics of the 4-pole, 1.5 kW squirrel-cage IM considered in this study, both for accounted and neglected SLLs [28].

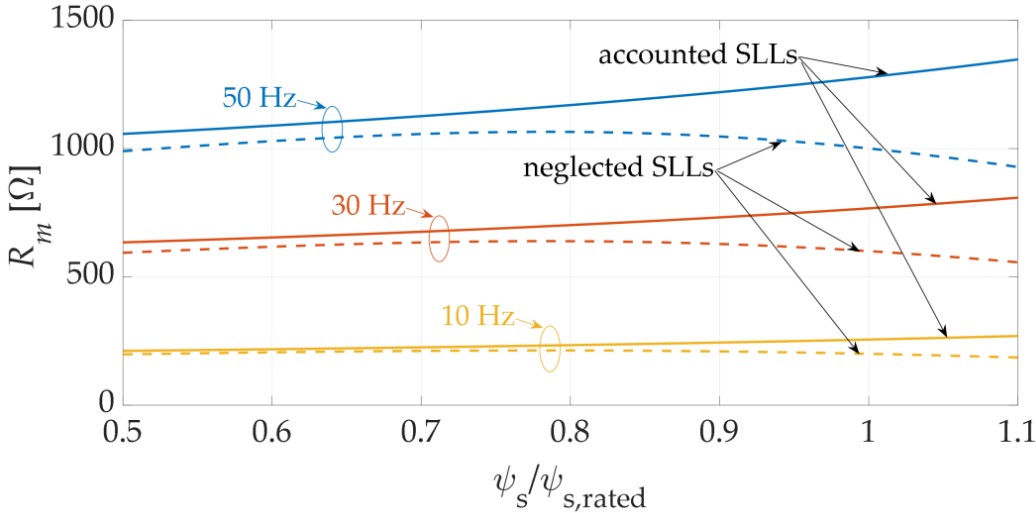

**Figure 2.** Iron-loss resistance characteristics of the considered IM.

Rated values of the considered IM are summarized in Table 1. Note that this IM was one of the four IMs used in [27] for experimental validation of the model in Figure 1a.

**Table 1.** Rated values of the tested IM.

| $P$ | $n$ | $\psi_r$ | $P_{Fe}$ | $P_{sll}$ | $R_s$ | $R_r$ | $L_{sl} = L_{rl}$ | $J$ |
|-----|-----|----------|----------|-----------|-------|-------|-------------------|-----|
| 1.5 kW | 1390 rpm | 0.864 Wb | 123.0 W | 70.7 W | 4.811 Ω | 3.154 Ω | 0.017 H | 0.003 kgm$^2$ |

### 2.2. Stray-Load Loss Modeling

SLLs are the portion of the IM's total losses not accounted for by the sum of friction and windage losses, stator and rotor winding losses, and iron losses. The corresponding resistance can be derived from the measurement data obtained from no-load and variable-load tests carried at different supply frequencies and stator flux magnitudes, as explained in [27]. Ultimately, the following formula describing the SLL resistance can be derived:

$$R_{sll}(\omega_e, \psi_s) = R_{sll-rated} \cdot \frac{\omega_e}{\omega_{e-rated}} \cdot \frac{\psi_s}{\psi_{s-rated}} \tag{10}$$

where $\psi_{s-rated}$ is the rated stator flux magnitude and $R_{sll-rated}$ is the $R_{sll}$ value at a rated frequency and magnetization (for the considered IM, $R_{sll-rated}$ = 1.8751 Ω).

Figure 3 shows the SLL resistance of the considered IM as being linearly dependent on both the stator flux and frequency [28].

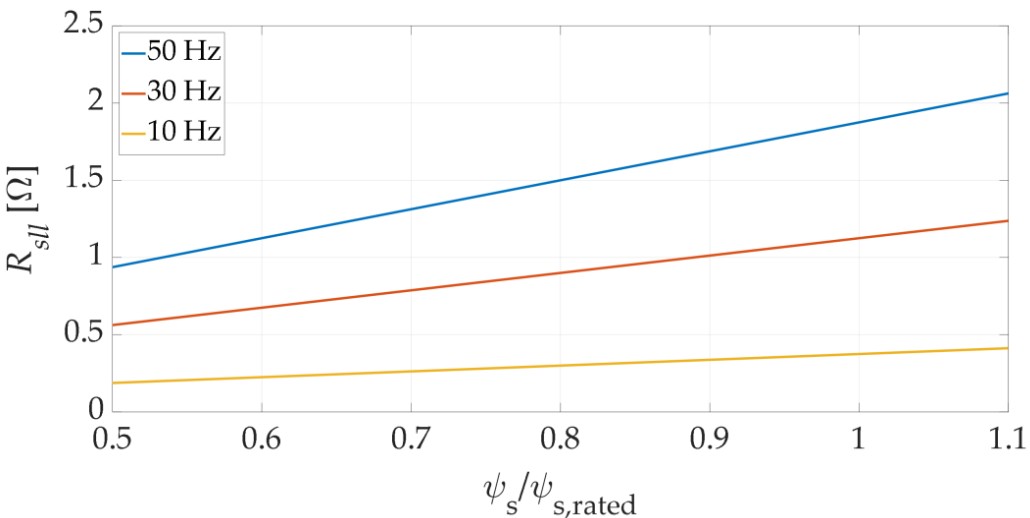

**Figure 3.** SLL resistance characteristics of the considered IM.

### 2.3. Magnetic Saturation Modeling

The magnetizing inductance dependency on the normalized stator flux magnitude can be derived from the standard no-load test. In the non-saturated region, $L_m$ is typically assumed to be constant—denoted by the dashed line in Figure 4—whereas in the saturated region, the obtained measurement points are approximated by a fitting curve—denoted by the solid line in Figure 4 [28].

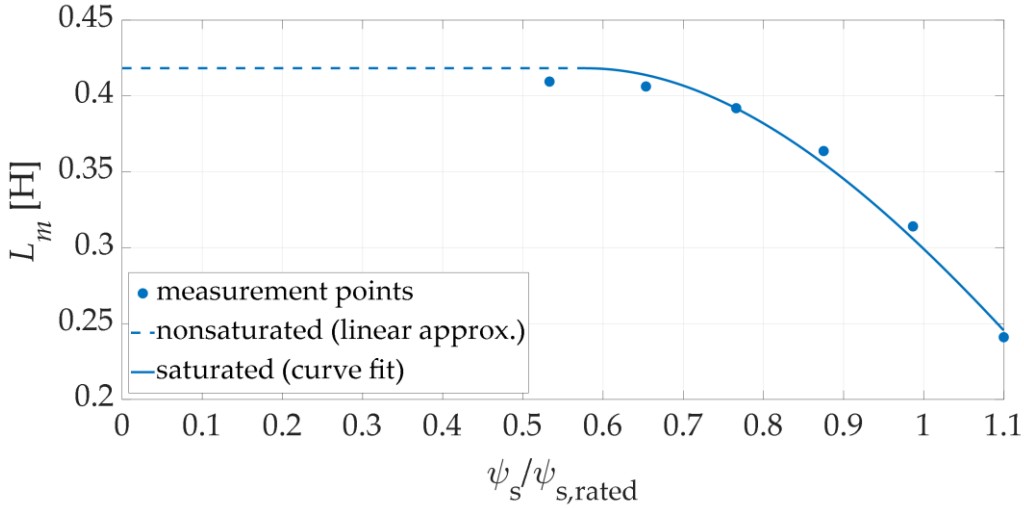

**Figure 4.** Magnetizing inductance characteristics of the considered IM.

It is important to note that the IM model shown in Figure 1a can be easily adapted to obtain models of different complexities. For example, by setting $L_m$ = const., $R_{sll}$ = 0, and $R_m$ → ∞, the conventional IM model is obtained; magnetic saturation can be accounted for by setting $L_m = f(\psi_s/\psi_{s\text{-}rated})$; by setting $R_{sll} \neq 0$, the model including the SLLs is obtained, etc. This fact is exploited in Section 5.1 to determine the recommended IM model's complexity for the proposed MPCC-based system.

Note here that the term "complexity" in this study primarily refers to the computational complexity of the IM model, which then has its repercussions on the computational complexity of the control algorithm based on that model. On the one hand, it implies the number of accounted phenomena present in the real IM, whereas on the other hand, it depends on the placement of the corresponding parameters within the equivalent circuit since it can affect the differential order of the model (a higher order implies a more complex

model). This placement must be physically sound but also allow a certain degree of free-dom (e.g., $R_m$ may be placed in parallel with $L_m$, or in series with $L_m$, or in parallel with $L_s$, as in Figure 1a). The complexity also depends on whether these parameters are represented as constant (less complex) or as variable (more complex), with the value depending on one or more variables. Lastly, this dependency can be linear (less complex) or described by mathematical functions of a higher order (more complex).

As already suggested, some assumptions were made in all the considered IM models to keep them computationally reasonable. These include the assumption of constant winding resistances, which in the real IM vary with respect to both the temperature and frequency (i.e., skin effect). Then, there is the assumption of constant leakage inductances, which are known to be flux-dependent, as well as the assumption of their even distribution between the stator and rotor side, which may not be the case in the real IM. Lastly, there are the already-mentioned assumptions of negligible eddy-current iron losses and shaft friction.

## 3. Control System Overview

In the proposed control system, the three-phase IM is supplied via the voltage source inverter (VSI), whose insulated-gate bipolar transistor (IGBT) switches are controlled by means of the DMPC. The proposed system's configuration is shown in Figure 5a, whereas the corresponding control algorithm, developed in the synchronously rotating *d-q* reference frame, is shown in Figure 5b (reference variables are marked with *). The MPCC algorithm in Figure 5b shares certain similarities with the indirect rotor-field-oriented (IRFO) control algorithm, shown in Figure 5c. For example, in both cases, the magnitude of the rotor flux space vector is assumed to be equal to its reference value; the reference *d*-axis component of the stator current is obtained by dividing the rotor flux reference with the magnetizing inductance, whereas the reference electromagnetic torque is obtained at the output of the rotor speed PI controller; the angle of the rotor flux space vector is used for the inverse Park transformation of the stator currents/voltages; both systems require measurement of the stator currents and rotor speed.

On the other hand, the proposed MPCC algorithm, as opposed to the IRFO, does not require two additional PI controllers for control of the *d*- and *q*-axis components of the stator current, which simplifies the system design significantly. Instead, the control of the stator phase currents is implemented within the MPC's cost function, as explained in Section 4. Also, the proposed system does not require an intermediate modulation stage to control the VSI's switches, nor does it require decoupling of the *d*- and *q*-axis components of the stator voltage. The decoupling terms are defined as follows:

$$v_{sTd-dec} = i^*_{sTq} \omega_e \sigma L_s \tag{11}$$

$$v_{sTq-dec} = i^*_{sTd} \omega_e \left( \sigma L_s + \frac{L_m^2}{L_r} \right) \tag{12}$$

where $\sigma = 1 - L_m^2/(L_s \cdot L_r)$ is the total leakage factor.

Note in Figure 5b,c that the $L_m$, $R_m$, and $R_{sll}$ values are expressed as a function of the rotor flux reference instead of the stator flux magnitude for the reasons explained in Section 5.1. Similarly, the $R_m$ and $R_{sll}$ values are expressed as a function of the rotor speed instead of the stator angular frequency.

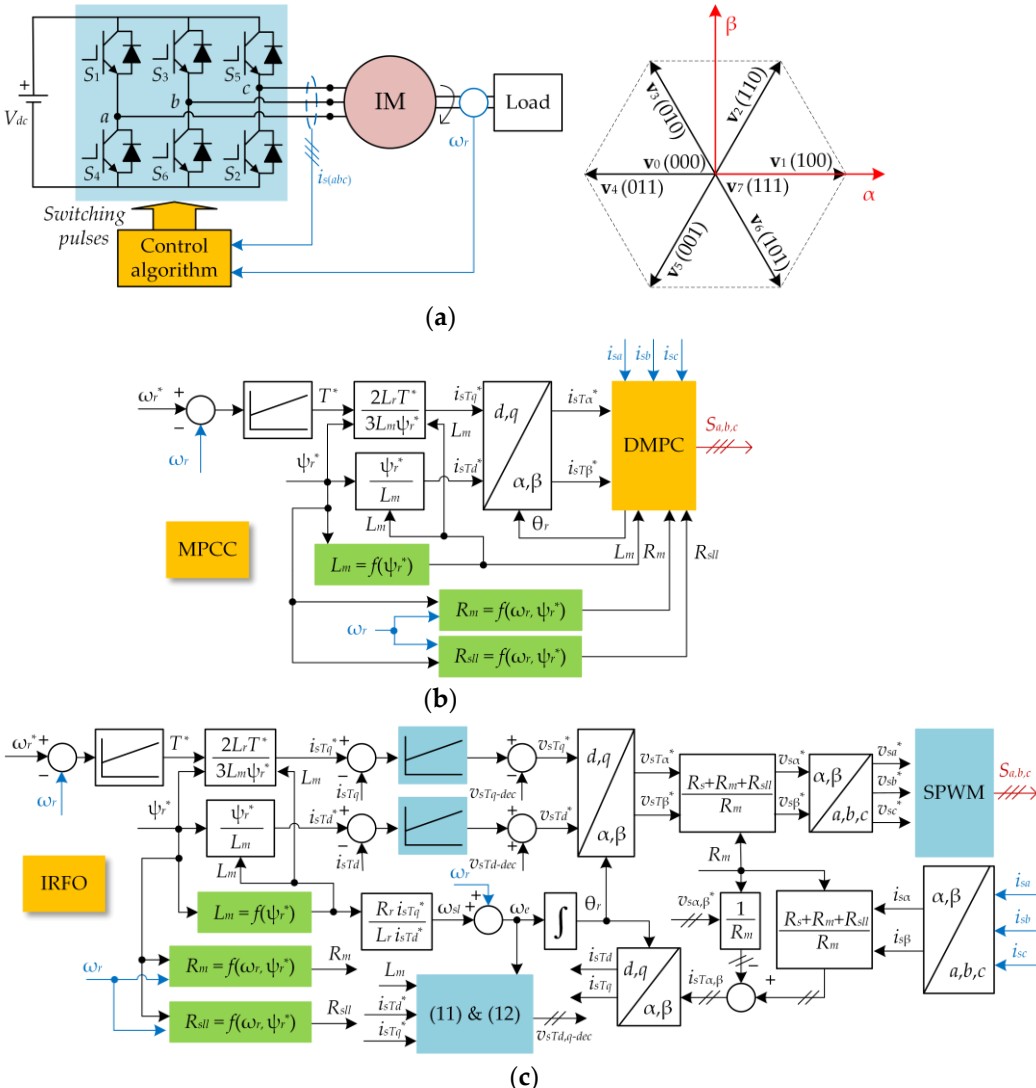

**Figure 5.** Considered IM control system: (**a**) system configuration and VSI voltage vectors, (**b**) proposed MPC algorithm, and (**c**) IRFO control algorithm.

## 4. Proposed Model Predictive Current Controller

By applying the forward Euler method to the equations of the IM model shown in Figure 1a (Appendix A), with $\omega_e = 0$, the discrete-time next-step prediction of the current $\mathbf{i}_{sT}$ from (8) can be obtained as follows (for the conventional IM model → $\mathbf{i}_{sT} = \mathbf{i}_s$, $\mathbf{v}_{sT} = \mathbf{v}_s$, and $R_{sT} = R_s$) [38]:

$$\mathbf{i}_{sT}(k+1) = \left(1 - \frac{T_s}{\tau_\sigma(k)}\right) \cdot \mathbf{i}_{sT}(k) + \frac{T_s}{\tau_\sigma(k)R_\sigma(k)} \cdot \left[k_r(k) \cdot \left(\frac{1}{\tau_r(k)} - j\omega_r(k)\right) \cdot \mathbf{\psi}_r(k) + \mathbf{v}_{sT}(k)\right] \tag{13}$$

where $T_s$ is the MPC sampling period, $\tau_\sigma = \sigma \cdot L_s/R_\sigma$, $R_\sigma = R_{sT} + R_r \cdot k_r{}^2$, $k_r = L_m/L_r$, and $\tau_r = L_r/R_r$.

Note in (13) that the parameters $\tau_\sigma$, $R_\sigma$, $k_r$, and $\tau_r$ are reevaluated in each time step, which allows $L_m$, $R_m$, and $R_{sll}$ to be modeled as non-constant parameters. This equation is solved for all eight possible switching combinations of the VSI, which, in turn, requires measurement of the stator current and the rotor angular speed, along with the rotor flux vector calculation (obtained from (A2) and (A4) and by applying the backward Euler discretization) as follows:

$$\boldsymbol{\psi}_r(k) = \boldsymbol{\psi}_r(k-1) \cdot \left(1 - \frac{T_s}{\tau_r(k)} + j\omega_r(k) \cdot T_s\right) + \mathbf{i}_{sT}(k) \cdot \frac{L_m(k)T_s}{\tau_r(k)} \tag{14}$$

From (14), the rotor flux angle required for the inverse Park transformation of the reference stator currents can be obtained as

$$\theta_r(k) = arctg\frac{\text{Im}(\boldsymbol{\psi}_r(k))}{\text{Re}(\boldsymbol{\psi}_r(k))} \tag{15}$$

In the field of power electronics, most MPC cost functions utilize the sum of the *absolute values* of the predicted tracking error components (i.e., the $\ell_1$-norm) because of the related computational simplicity [13,39–43]. However, as it was shown in [8,44], this may lead to closed-loop instability and performance deterioration. On the other hand, using the sum of *squares* of the tracking error components (i.e., the $\ell_2$-norm) guarantees closed-loop stability, good tracking performance, and low distortions, especially when CEP is also implemented. Due to this reason, the proposed MPCC utilizes the $\ell_2$-norm cost function as follows:

$$g = (i^*_{sT\alpha}(k) - i_{sT\alpha}(k+1))^2 + \left(i^*_{sT\beta}(k) - i_{sT\beta}(k+1)\right)^2 \tag{16}$$

where $i_{s\alpha}(k+1)$ and $i_{s\beta}(k+1)$ are the stationary α-β components of the predicted stator current space vector $\mathbf{i}_{sT}(k+1)$; note that, for simplicity, the reference currents from the $(k+1)$-th step are approximated by those from the $k$-th step.

Furthermore, most direct MPCs reported in the literature consider a one-step prediction horizon and exclude the CEP, which then resembles deadbeat control [45]. However, deadbeat controllers are known to be sensitive to model mismatches and parameter uncertainties. In addition, the switching frequency is, in this case, limited only by the MPC sampling period as $f_{sw} < 1/(2T_s)$. It was shown in [8] that under such conditions, the DMPC does not outperform conventional PWM methods in terms of current distortions. To ensure a desirable steady-state performance, a high sampling-to-switching frequency ratio is required. This may be achieved by reducing the switching frequency through penalization of the control effort. In this study, the CEP is implemented by constraining simultaneous switching to two VSI branches ($f_{sw} < 1/(3T_s)$)—by means of the parameter $h_{sw}$ included in the cost function—and by including the weighted switching penalization term $\lambda_{sw} \cdot n_{sw}$ in the cost function as follows:

$$g = \left(i^*_{sT\alpha}(k) - i_{sT\alpha}(k+1)\right)^2 + \left(i^*_{sT\beta}(k) - i_{sT\beta}(k+1)\right)^2 + h_{sw} + \lambda_{sw} \cdot n_{sw}$$
$$n_{sw} = |S_a(k) - S_a(k-1)| + |S_b(k) - S_b(k-1)| + |S_c(k) - S_c(k-1)| \tag{17}$$
$$\text{if } n_{sw} = 3, h_{sw} = 10^{10}; \text{ else } h_{sw} = 0$$

where $n_{sw}$ equals the number of VSI branches with simultaneous switching and $\lambda_{sw}$ is the adjustable weighting factor.

The flowchart of the proposed DMPC algorithm is presented in Figure 6.

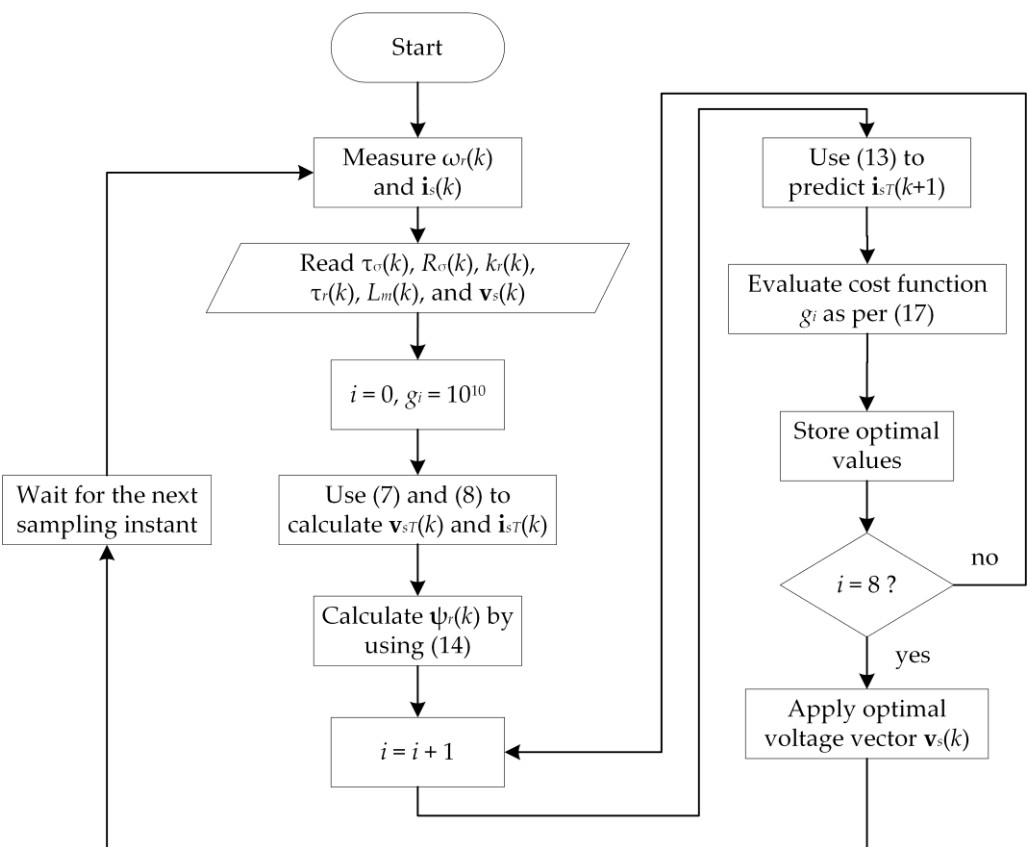

**Figure 6.** Flowchart of the proposed DMPC algorithm.

### 5. Results and Discussion

An extensive simulation analysis is first carried out in the MATLAB Simulink environment using several MPC algorithms based on IM models of different complexities to determine the best overall candidate for the proposed MPCC system. The MPC algorithm based on thus selected IM model is then utilized for the assessment of the impact of the CEP on the system's performance. Finally, the steady-state and dynamic performance of the proposed MPCC system is compared to that of the IRFO system (Figure 5c). In all simulations, a variable-step solver (ode45, Dormand-Prince) with the maximum step size set to 1 μs was used for simulating the IM and the VSI. Other solver parameters were set to default values.

Besides the assumptions related to the IM parameters, discussed in Section 2, other assumptions made in the simulation model include the following:

- Both DC-link voltage and load torque are assumed constant.
- The inverter switches are assumed ideal.
- Feedback signals do not contain noise, offset, or gain error.
- There is no electromagnetic interference (EMI).
- There is no delay in application of the command (switching) signal.

### 5.1. MPCC Performance with Different IM Models

The IM models which are in this subsection considered for the MPC design are all somewhat simplified versions of the IM model in Figure 1a, starting from the simplest to the more complex ones. They differ with respect to the parameter settings as follows:

(a)　$L_m = L_{m\text{-}nonsat}$, $R_{sll} = 0$, and $R_m \to \infty$
(b)　$L_m = f(\psi_r^*)$, $R_{sll} = 0$, and $R_m \to \infty$
(c)　$L_m = f(\psi_r^*)$, $R_{sll} = 0$, and $R_m = R_{m\text{-}rated}$
(d)　$L_m = f(\psi_r^*)$, $R_{sll} = 0$, and $R_m = f(\omega_r)$

(e)    $L_m = f(\psi_r^*)$, $R_{sll} = f(\omega_r, \psi_r^*)$, and $R_m = f(\omega_r, \psi_r^*)$

The model variant $a$ is, in fact, the conventional IM model shown in Figure 1c, in which the magnetic saturation, the iron losses, and the SLLs are entirely ignored. The magnetizing inductance is set equal to the corresponding non-saturated value, as in [17,23], the SLL resistance is set to zero, and the iron-loss resistance is set to $10^{10}$ Ω for practical reasons. The model variant $b$ includes the magnetic saturation, as is the case with all the remaining variants $c$-$e$. One may notice, however, that the magnetizing inductance's dependency on the (normalized) *stator flux magnitude* (Figure 4) is here approximated by the dependency on the (normalized) *rotor flux reference* (i.e., $\psi_s/\psi_{s\text{-}rated} \rightarrow \psi_r^*/\psi_{r\text{-}rated}$). This is because $\psi_r^*$ is readily available in the control algorithm, whereas $\psi_s$ would have to be additionally calculated, thus increasing the MPC's computational burden. It is assumed that $\psi_s/\psi_{s\text{-}rated} \approx \psi_r^*/\psi_{r\text{-}rated}$. The model variant $c$ additionally includes the iron losses by means of the constant iron-loss resistance whose value corresponds to the rated operating conditions. In the model variant $d$, the iron-loss resistance is assumed to be proportional to the rotor speed as $R_m = R_{m\text{-}rated} \cdot \omega_r/\omega_{r\text{-}rated}$. Again, one may notice that in Section 2.1, $R_m$ was defined as being proportional to the (normalized) stator angular frequency, whereas here, the (normalized) angular rotor speed $\omega_r$ is utilized instead for practical reasons (i.e., $\omega_e/\omega_{e\text{-}rated} \rightarrow \omega_r/\omega_{r\text{-}rated}$). This is because $\omega_r$ is readily available in the control algorithm, whereas $\omega_e$ would have to be additionally calculated, and $\omega_e/\omega_{e\text{-}rated} \approx \omega_r/\omega_{r\text{-}rated}$ holds true in normal operation. Lastly, in the most elaborate model variant $e$, $R_m$, and $R_{sll}$ are linearly dependent on both the rotor flux reference and angular rotor speed in a way similar to that described by (9) and (10), but with the following substitutions applied: $\psi_s/\psi_{s\text{-}rated} \rightarrow \psi_r^*/\psi_{r\text{-}rated}$ and $\omega_e/\omega_{e\text{-}rated} \rightarrow \omega_r/\omega_{r\text{-}rated}$.

The IM model presented in Figure 1a is taken as the reference model in this study and is utilized to simulate the *actual* induction machine (i.e., the one *outside* of the MPC algorithm). The sampling period for the MPC was set to $T_s$ = 20 μs ($f_{sw\text{-}max}$ = 25 kHz), whereas the sampling period for the rest of the control algorithm was set to $T_{s1}$ = 1 ms to reduce the computational burden while not sacrificing much the performance. The DC-link voltage was in all simulations set to 520 V. The rotor flux reference was set to the corresponding rated value ($\psi_r^*$ = $\psi_{r\text{-}rated}$). Consequently, in all the considered model variants except for the variant $a$, the $L_m$ value in the MPC algorithm was, in fact, constant and equal to 0.2991 H (i.e., the value corresponding to the rated stator flux in Figure 4). Hence, the model variant $b$ is in this particular case equivalent to the conventional IM model in which the $L_m$ value is fixed at the corresponding rated value. At the same time, the $L_m$ value in the IM model *outside* of the MPC algorithm may differ depending on the actual magnetizing conditions (i.e., the difference between the reference and actual rotor flux). In general, the closer the rotor flux reference corresponds to the actual rotor flux magnitude, the closer the $L_m$ value in the MPC algorithm corresponds to the actual $L_m$ value, which depends on the accuracy of the IM model utilized for the MPC. The dependency of the $L_m$ (as well as the $R_m$) value on the IM operating flux would have been more pronounced in the case of variable rotor flux reference (e.g., in flux-weakening operation above rated speed or in certain IM loss-minimization strategies), but such analysis falls out of the scope of this study. In any case, the MPC algorithm in Figure 5b allows us to consider such variations if required.

First, the cost function defined in (16) is utilized and the steady-state values of the following performance metrics are considered: the total harmonic distortion of the stator current $THD_I$ (related with the stator current error and IM harmonic losses), the average switching frequency $f_{sw\text{-}avg}$ (related with the control effort and power converter losses), the rotor flux magnitude error $\Delta\psi_r$ (i.e., the ratio of the actual and the reference magnitude of the rotor flux vector), the rotor flux angle error $\Delta\theta_r$ (i.e., the angular misalignment between the actual and the reference rotor flux vector), and the product $THD_I \cdot f_{sw\text{-}avg}$. These are all presented in Figures 7–11 as a function of the normalized load torque and rotor speed in the form of 3D surface plots. Note, however, that the plots obtained for the model variant $e$ are very similar to those obtained for the variant $d$, so only the corresponding *median* and *mean*

values are provided in the brackets in Figures 7d, 8d, 9d, 10d and 11d for comparison (the corresponding 3D surface plots are provided as supplementary .tif files). This similarity is also evident from the results given in Table 2.

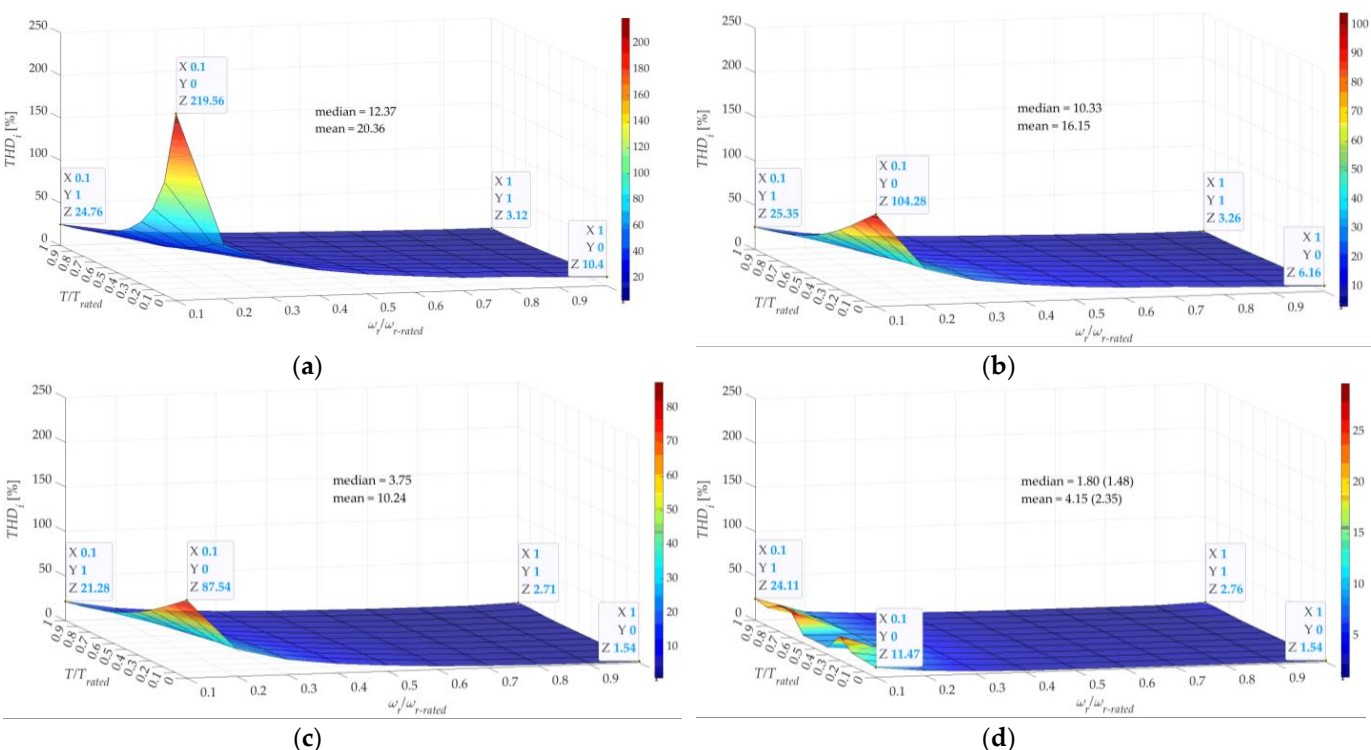

**Figure 7.** Steady-state values of the total harmonic distortion of the stator current obtained for different IM model variants: (**a**) variant *a*, (**b**) variant *b*, (**c**) variant *c*, and (**d**) variant *d*.

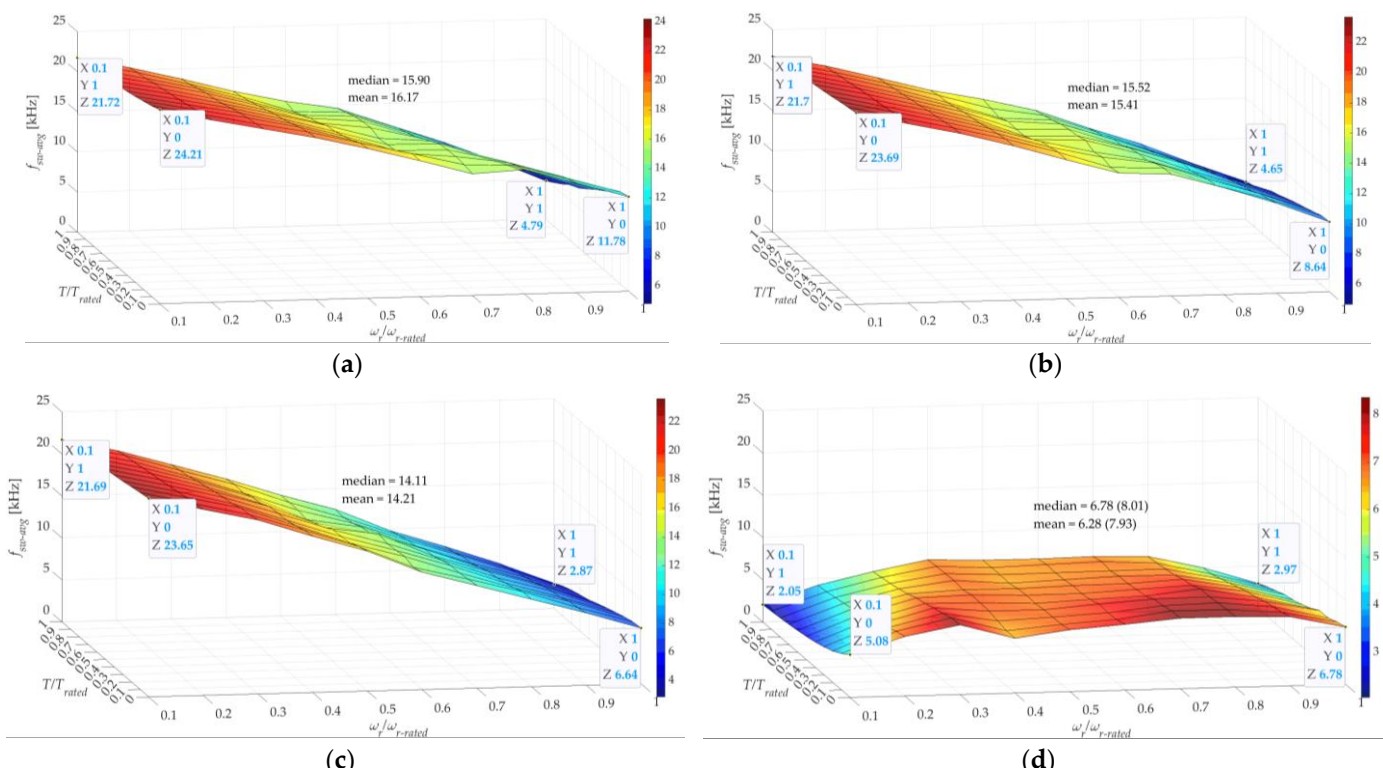

**Figure 8.** Steady-state values of the average switching frequency obtained for different IM model variants: (**a**) variant *a*, (**b**) variant *b*, (**c**) variant *c*, and (**d**) variant *d*.

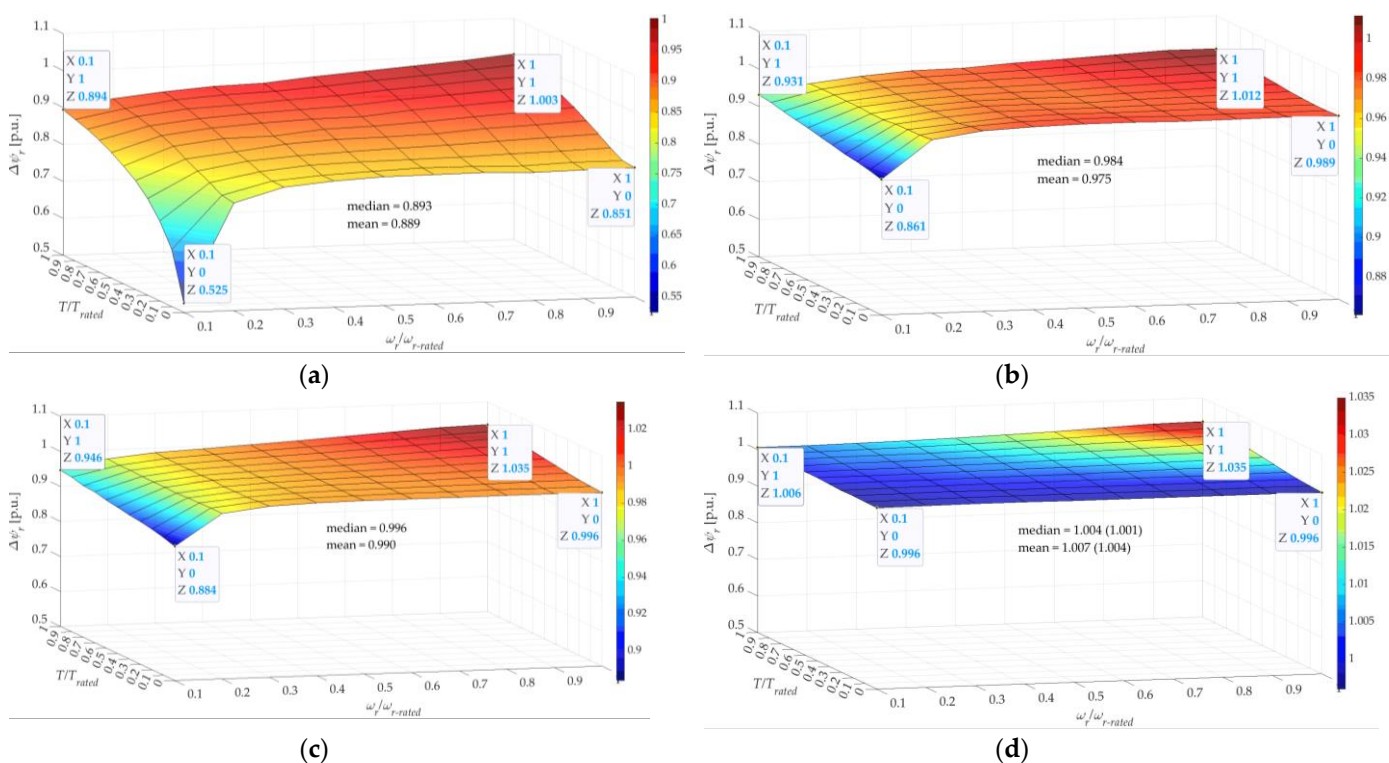

**Figure 9.** Steady-state values of the rotor flux magnitude error obtained for different IM models: (**a**) variant *a*, (**b**) variant *b*, (**c**) variant *c*, and (**d**) variant *d*.

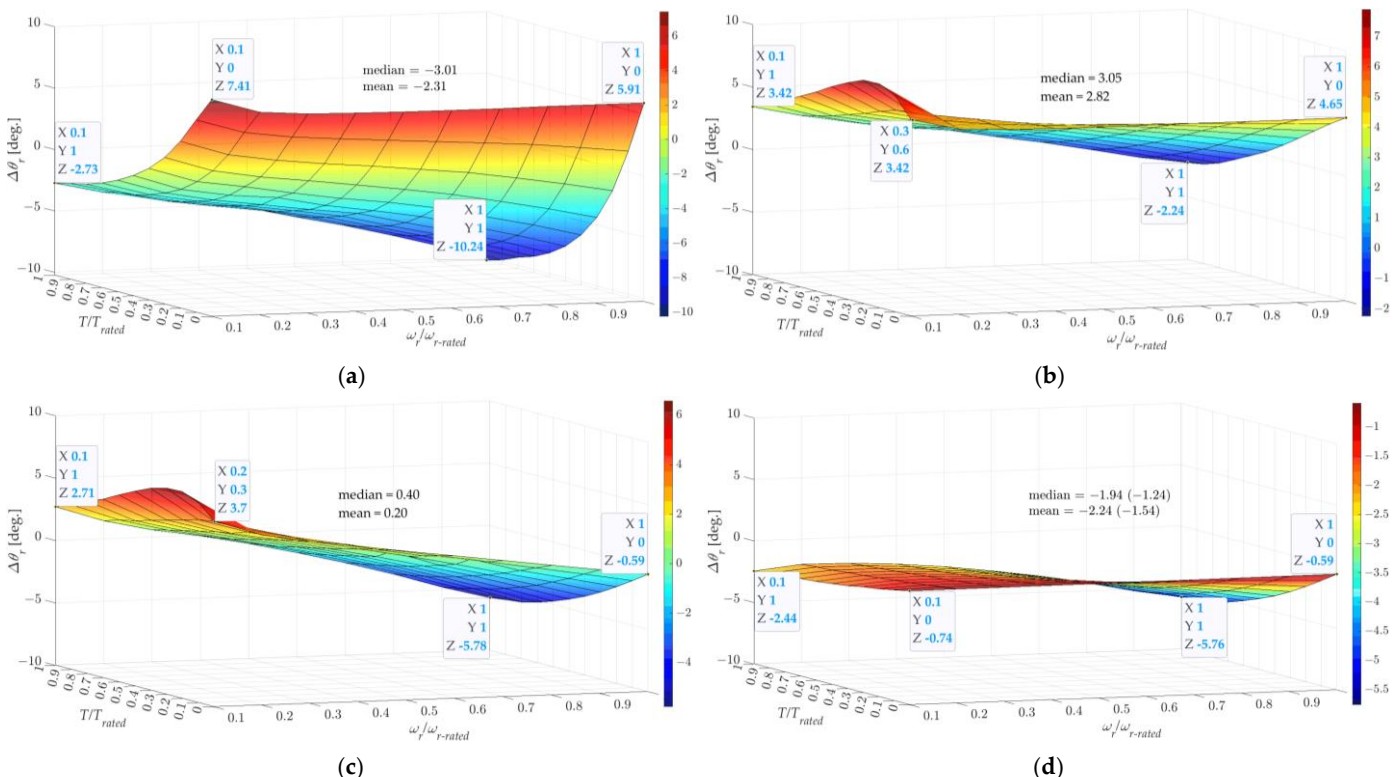

**Figure 10.** Steady-state values of the rotor flux angle error obtained for different IM models: (**a**) variant *a*, (**b**) variant *b*, (**c**) variant *c*, and (**d**) variant *d*.

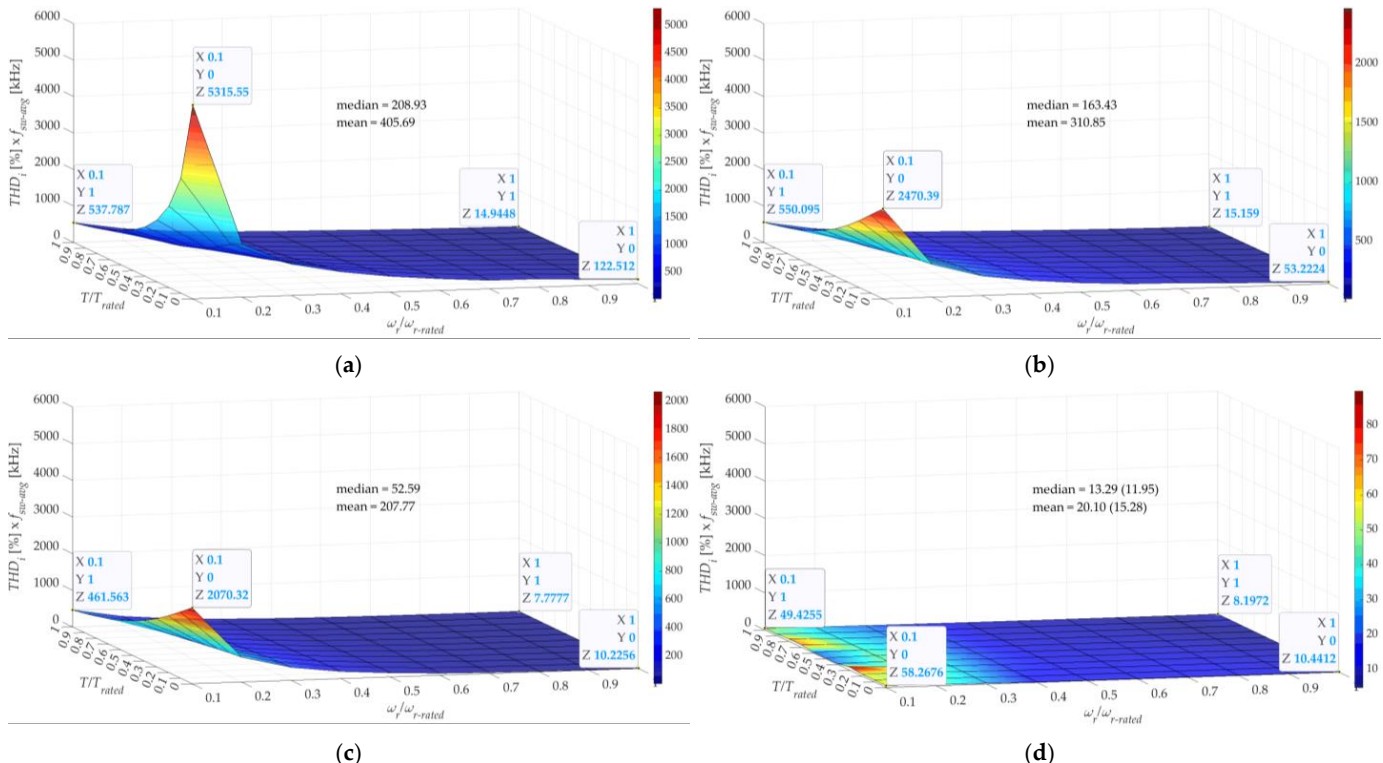

**Figure 11.** Steady-state values of the product $THD_I \cdot f_{sw\text{-}avg}$ obtained for different IM models: (**a**) variant *a*, (**b**) variant *b*, (**c**) variant *c*, and (**d**) variant *d*.

**Table 2.** Percentage shares of the operating points within given margins of the performance metrics.

| IM Model | $THD_I$ ($\leq$5%) | $f_{sw\text{-}avg}$ ($\leq$10 kHz) | $\Delta\psi_r$ (1 $\pm$ 0.02 p.u.) | $\Delta\theta_r$ (0 $\pm$ 2 deg.) |
|---|---|---|---|---|
| *a* | 10.91% | 11.82% | 4.55% | 20.91% |
| *b* | 14.55% | 19.09% | 67.27% | 30.00% |
| *c* | 56.36% | 29.09% | 72.73% | 50.91% |
| *d* | 80.00% | 100.00% | 90.00% | 52.73% |
| *e* | 91.82% | 78.18% | 92.73% | 72.73% |

Each of the plots in Figures 7–11 comprise 110 operating points, encompassing base regions of rotor speed and load torque. Table 2 shows the percentage shares of the operating points that are within the given margins of the considered performance metrics.

It is evident from the presented plots and numerical results that inclusion of the magnetic saturation in the MPC model as in the model variant *b* significantly reduces the rotor flux magnitude error, whereas its contribution to the improvement of the other considered performance metrics is more modest, but also worth noting.

The inclusion of iron losses, as in the model variant *c*, contributes the most to the reduction in the THD of the stator currents and, thus, to the reduction in the IM harmonic losses, but it also improves all the other considered performance metrics, resulting in a considerable reduction in the product $THD_I \cdot f_{sw\text{-}avg}$.

By setting the iron-loss resistance as proportional to the rotor speed, as in the model variant *d*, all the considered performance metrics are additionally enhanced, especially in the low-speed/low-torque regions. In low-torque IM operation, the share of the iron losses becomes more pronounced due to the reduction in both the winding losses and the SLLs, so it becomes increasingly important to take them into account. Similarly, in the model variant *c*, as opposed to the model variant *d*, the $R_m$ value is significantly overestimated (i.e., the iron losses are underestimated) in the low-speed operating region because the $R_{m\text{-}rated}$ value



is determined for the rated frequency of 50 Hz. The model variant *d* enables a more accurate prediction of the stator currents in this region and, thus, better selection of the optimal voltage vector. Consequently, with the model variant *d*, the average switching frequency is decreased by more than two times, with the highest corresponding value reaching only 8.35 kHz, as compared to 23.65 kHz previously obtained for the model variant *c*. This inevitably reduces the converter losses. In addition to that, 80% of the considered operating points satisfy the criterion $THD_I \leq \pm 5\%$. Note also that even though the *median* and *mean* values of the rotor flux angular error are slightly greater than those obtained for the model variant *c*, the corresponding range is reduced by half. The largest overall decrease was recorded in the product $THD_I \cdot f_{sw-avg}$, with the corresponding *mean* value decreased more than tenfold.

The implementation of the most elaborate model variant *e* somewhat noticeably reduces the stator currents' THD and the rotor flux angular error. However, this is achieved at the cost of an increase in the average switching frequency, so the product $THD_I \cdot f_{sw-avg}$ is only marginally reduced ($f_{sw-avg}$ does not surpass 12 kHz at any point). Given the rather modest improvements in the performance metrics due to the application of the model variant *e*, it seems difficult to justify using this IM model for the MPCC design considering the increased computational cost and more complicated procedure of obtaining model parameters.

Summed up, the IM model variant *d*, i.e., the one including the magnetic saturation and rotor speed-dependent iron-loss resistance, is recommended for the proposed MPCC. Anything more complex than that does not result in substantial performance improvements while implying a more complex process of determining model parameters and increased computational burden, whereas less complex IM model variants result in considerable performance degradation. Hence, the recommended model variant is utilized for the remaining analysis.

### 5.2. Impact of Control Effort Penalization

As explained in Section 4, the control effort in this study is penalized in two ways: by constraining the number of simultaneous switching transitions to a maximum of two VSI branches ($f_{sw-max}$ = 16.67 kHz) and by introducing the weighted switching penalization term in the cost function. To evaluate its impact, the weighting factor $\lambda_{sw}$ in this study is set to 0.05. This value could have been further optimized, e.g., through minimization of the product $THD_I \cdot f_{sw-avg}$, but such analysis falls out of the scope of this study. The simulation parameters, the MPC code, and the simulation model segments in MATLAB-Simulink of the MPCC system based on the IM model variant *d* and including the CEP are all provided as supplementary .pdf files.

The obtained results are presented in Figure 12 and in Table 3. Note that the *median* and *mean* values in the brackets in Figure 12 refer to the results obtained without the CEP.

It is evident from the above results that the average switching frequency can be further considerably reduced by implementing the CEP, while not sacrificing too much the other considered performance metrics. In fact, only the share of the operating points that meets the condition $THD_I \leq 5\%$ has decreased by about 10%. This may seem significant, but if the $THD_I$ margin had been set only 1% higher, the share would be equal to 80% in both cases—with and without the CEP. As for the rotor flux magnitude and angle errors, the corresponding plots given in Figure 12c,d, respectively, are practically indistinguishable. It is also interesting to note that the achieved reduction in the product $THD_I \cdot f_{sw-avg}$ is approximately the same as it was due to the application of the most elaborate IM model in Section 5.1. It is also possible that this performance metric could be further reduced by optimizing $\lambda_{sw}$.

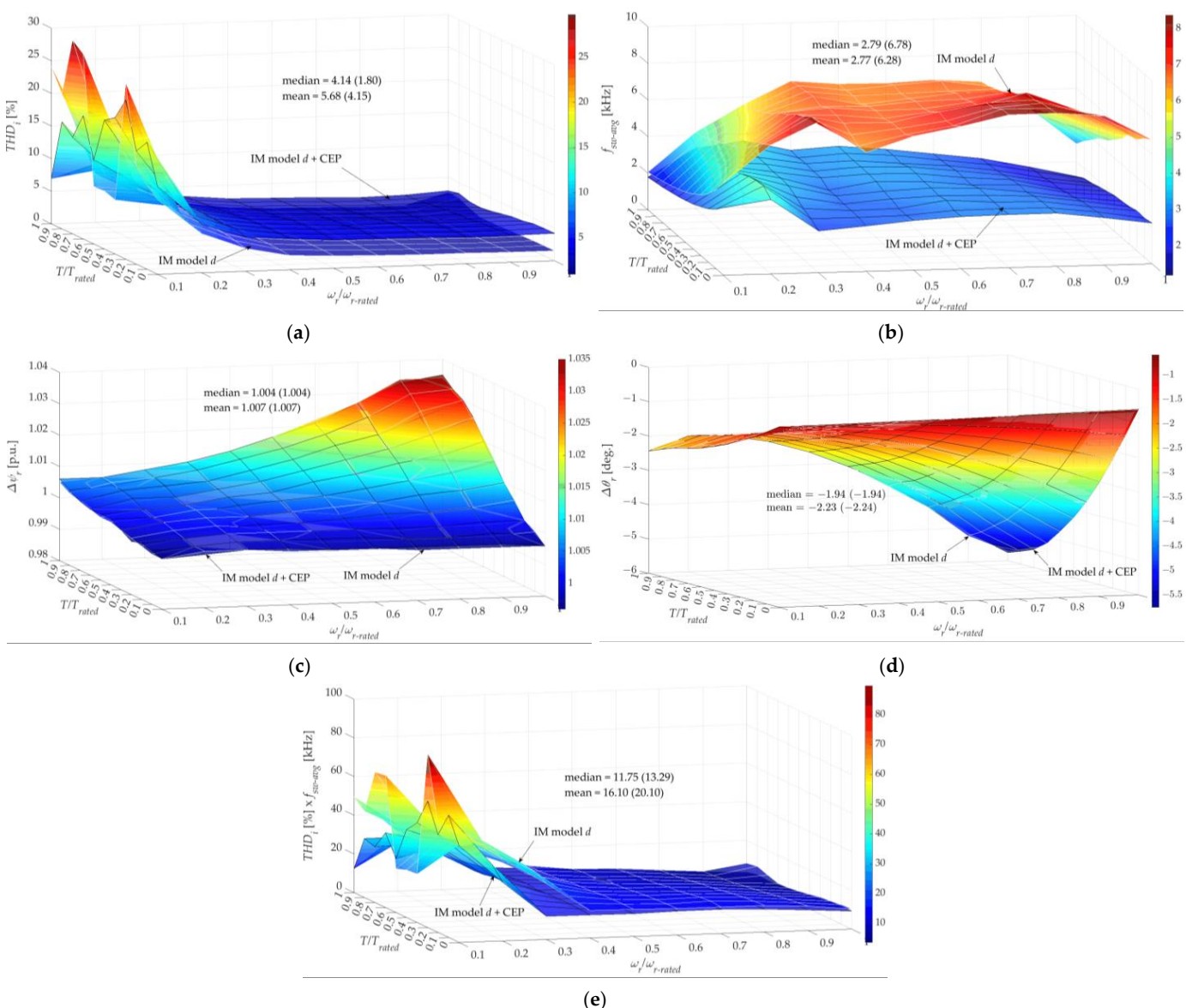

**Figure 12.** Steady-state values of the performance metrics obtained for the model variant *d*—with and without CEP: (**a**) total harmonic distortion of the stator current, (**b**) average switching frequency, (**c**) rotor flux magnitude error, (**d**) rotor flux angle error, and (**e**) product $THD_I \cdot f_{sw\text{-}avg}$.

**Table 3.** Percentage shares of the operating points within given margins of the considered performance metrics—CEP with $\lambda_{sw} = 0.05$.

| IM Model | $THD_I$ ($\leq$5%) | $f_{sw\text{-}avg}$ ($\leq$5 kHz) | $\Delta\psi_r$ ($1 \pm 0.02$ p.u.) | $\Delta\theta_r$ ($0 \pm 2$ deg.) |
|---|---|---|---|---|
| *d* | 80.00% | 20.00% | 90.00% | 52.73% |
| *d* + CEP | 70.91% | 100.00% | 89.09% | 52.73% |

### 5.3. MPCC vs. IRFO Performance Comparison

In this section, the performance of the proposed MPCC-based system (Figure 5b) is compared to that of the IRFO-based system (Figure 5c). As already explained, these two systems share many similarities up to the point of acquiring the reference stator currents, but they fundamentally differ in how these currents are further handled to eventually obtain the switching signals for the VSI. One of the differences is that the IRFO system uses a sinusoidal PWM modulator with a fixed switching frequency, whereas in the

case of the MPCC system, the switching frequency varies. To enable as fair a comparison as possible, the switching frequency of the PWM modulator was adjusted for each considered operating point and set equal to the average switching frequency of the MPCC algorithm (Figure 12b). The same parameter values were used for the speed PI controller in both systems, and these were tuned through a trial-and-error procedure. In both systems, the sampling period $T_{s1}$ = 1 ms was used for the part dedicated to acquiring the reference stator currents and calculation of IM model parameters. The rest of the MPCC algorithm, including the model predictive controller and the inverse Park transformation of the stator currents, was executed with the sampling period $T_s$ = 20 μs, which gives a sampling frequency ratio of 50. However, according to [46], for cascade speed control systems, such is the IRFO, and for switching frequencies below 30 kHz, the sampling frequency of the speed control loop should be 2–10 times lower than the main sampling frequency. Therefore, in the considered IRFO algorithm, the main sampling frequency was reduced to 10 kHz to achieve a sampling frequency ratio of 10, which is the recommended value closest to the ratio implemented for the MPCC. In the case of the IRFO algorithm, this part of the algorithm includes two internal (current) control loops, with the respective PI controllers and the decoupling and inverse Park transformation of the stator voltages. The current PI controllers' parameters were tuned according to the procedure described in Appendix C [47]. Both control algorithms are designed based on the equations of the IM model variant *d*, described in Section 5.1. The CEP was additionally implemented in the case of the MPCC, as explained in Section 4, with $\lambda_{sw}$ set to 0.05 as in Section 5.1.

The reference speed and load variations during the simulation are summarized in Table 4, whereas the obtained dynamic responses are presented in Figure 13. The rotor flux reference was set to $\psi_r^* = \psi_{r\text{-}rated}$, whereas the DC-link voltage was set to 520 V. The stator current THD values are additionally provided in Table 5.

It is fair to say that both considered algorithms provide very similar performances with the selected controller settings. The proposed MPCC algorithm in most cases (except for the initial run up) ensures a slightly smaller overshoot in the rotor speed. This is particularly evident at *t* = 0.5 s when the speed reference signal was increased in a step manner from 0.5 to 1.0 p.u., resulting in an overshoot reduction from 23% to only 8%. The IRFO algorithm, on the other hand, ensures a slightly smaller settling time in most cases, whereas the rise time is almost identical. As for the torque response, the MPCC algorithm ensures lower ripple at higher rotor speeds, regardless of the load (*t* = 0.5–2 s), but it also induces higher ripple at reduced rotor speeds in combination with high loads (*t* = 2.5–3 s). Lower overshoots accompanied with fewer oscillations are observed in the rotor flux response when the MPCC is used. On the other hand, the actual rotor flux magnitude corresponds better to its reference value under rated load and speed conditions when the IRFO is applied (*t* = 1–1.5 s). Lastly, at higher rotor speeds, the stator current THD values are considerably lower for the proposed MPCC algorithm, regardless of the load (*t* = 0.5–2 s), which results in lower harmonic losses of the IM. However, as the rotor speed decreases below half the rated value, the IRFO algorithm gets increasingly better in this regard, compared to the proposed MPCC algorithm (*t* = 2–4 s).

**Table 4.** Reference rotor speed and load torque variations during simulation.

| *t* [s] | 0–0.5 | 0.5–1 | 1–1.5 | 1.5–2 | 2–2.5 | 2.5–3 | 3–3.5 | 3.5–4 |
|---|---|---|---|---|---|---|---|---|
| $\omega_r^*$ [p.u.] | 0.5 | 1.0 | 1.0 | 1.0 | 0.25 | 0.25 | 0.25 | −0.25 |
| *T* [p.u.] | 0.0 | 0.0 | 1.0 | 0.0 | 0.0 | 1.0 | 0.0 | 0.0 |

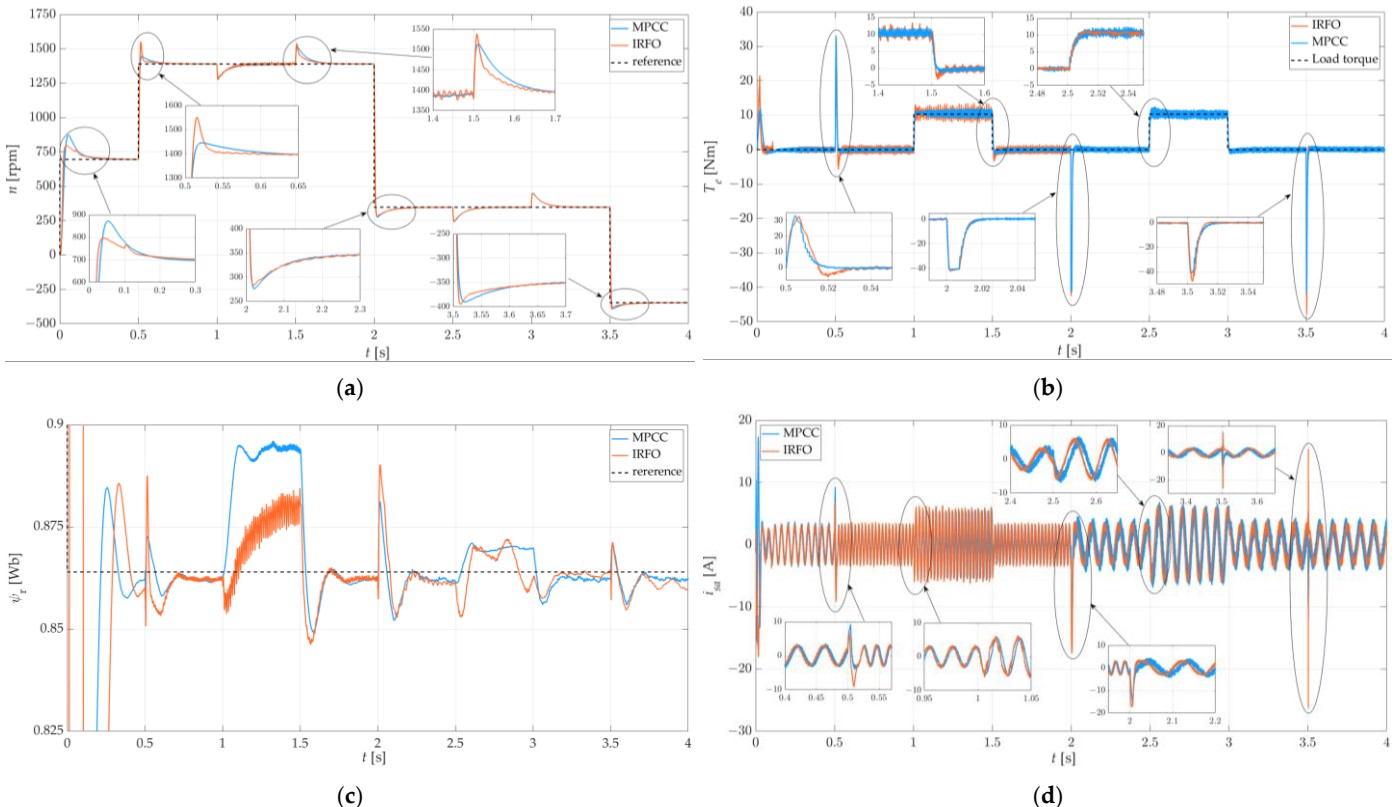

**Figure 13.** Dynamic responses recorded for the proposed MPCC algorithm and the IRFO algorithm: (**a**) rotor speed, (**b**) electromagnetic torque, (**c**) rotor flux magnitude, and (**d**) stator phase current (phase *a*).

**Table 5.** Stator current THD values obtained for the proposed MPCC-based system and the IRFO system.

| $t$ [s] | 0–0.5 | 0.5–1 | 1–1.5 | 1.5–2 | 2–2.5 | 2.5–3 | 3–3.5 | 3.5–4 |
|---|---|---|---|---|---|---|---|---|
| THD [%] (MPCC) | 9.15 | 5.65 | 3.13 | 5.67 | 20.79 | 12.72 | 19.69 | 19.69 |
| THD [%] (IRFO) | 8.56 | 8.89 | 8.13 | 9.03 | 11.99 | 6.62 | 12.12 | 12.12 |

To evaluate the computational cost of the considered control algorithms, the corresponding execution times (ETs) were determined for the previously described simulation by using the *tic toc* routine in the MATLAB Simulink (R2022b). For each of the considered algorithms, the average ET value obtained for ten consecutively executed simulations was determined. In addition to the obtained ET values, the results presented in Table 6 include percentage differences in the ET values of the MPCC-based models compared to the IRFO-based model. The letters *a–e* in the MPCC algorithm labels in Table 6 denote the utilized IM model variant. For this analysis, a PC with the following characteristics was used: Intel(R) Core(TM) i9-12900, 2.40 GHz, 16.0 GB DDR5.

**Table 6.** Simulation execution times of the considered control algorithms.

| Control Algorithm | IRFO | MPCC-*a* | MPCC-*b* | MPCC-*c* | MPCC-*d* | MPCC-*e* | MPCC-*d* + *CEP* |
|---|---|---|---|---|---|---|---|
| ET [s] | 16.14 | 15.74 | 15.75 | 15.84 | 15.90 | 15.93 | 16.36 |
| ΔET [%] | - | −2.83 | −2.42 | −1.86 | −1.49 | −1.30 | +1.36 |

Regardless of the utilized IM model variant, the application of the MPCC algorithm decreases the ET value by a few percentage points compared to the IRFO algorithm. The lowest ET was expectedly recorded for the MPCC based on the simplest IM model (MPCC-*a*), whereas only a minor gradual increase in the ET value was recorded when more complex MPCC algorithms were utilized. It is only when the CEP is additionally implemented (MPCC-*d* + CEP) that the MPCC algorithm results in a slightly higher ET value compared the IRFO algorithm.

### 5.4. Performance Comparison with the Existing Competing MPC Algorithms

The only two MPC algorithms from the literature that include IM magnetic saturation and iron losses and are, hence, to some extent comparable to the one proposed here are those proposed in [21,22].

The IM model considered in [21] involves a computationally more complex representation of the iron-loss resistance and IM inductances; namely, the iron-loss resistance is represented as a function of the magnetizing flux magnitude and its time derivative, whereas the magnetizing inductance and the stator and rotor leakage inductances—both static and dynamic—are represented as a function of the magnetizing and leakage flux magnitudes. A PWM modulator is additionally required in [21] to ensure constant switching frequencies, whereas the cost function includes absolute errors in the predicted rotor flux space-vector components, so it does not fall into the MPCC group. In addition, as already mentioned, the utilized $\ell_1$-norm does not guarantee closed-loop stability The simulation responses obtained for a 4-pole 2.2 kW IM are very similar to those obtained for the proposed MPCC in terms of the observed speed and torque transients and steady-state errors in the rotor flux magnitude. However, a lack of available data (e.g., switching frequency, overshoots, settling times, ripple, THD, etc.) prevents a better comparison.

The cost function considered in [22] includes the absolute differences (i.e., $\ell_1$-norm) of the rotor speed and flux, whereas the IM parameters, including the iron-loss resistance and the magnetizing inductance, are updated online by using an optimized GA. This in itself contributes to a non-negligible increase in computational complexity. The experimental validation involved a 4-pole 1.5 kW IM, but it only included dynamic responses of the electromagnetic torque, rotor speed, and stator phase current to speed reference step-changes from 1 p.u. to 0.5 p.u. and back, all recorded for the rated load and unspecified rotor flux reference. The speed response is notably slower with larger overshoots compared to that in Figure 13, but the torque overshoots seem to be smaller. Again, a lack of available data (e.g., control system diagram, rotor flux reference value, switching frequency, sampling period, THD, ripple, etc.) and a very limited number of presented results prevents a better comparison.

### 5.5. Practical Considerations and Challenges

Many of the practical considerations and challenges are related to the assumptions made in the simulation model and in the IM model itself, so they are to be dealt with in future studies. Some examples are given below.

The impact of the measurement noise on MPC is addressed in very few studies [48,49]. Given the fact that the MPCC performance largely depends on the quality of the measured current and speed signals, this issue should not be overlooked and some kind of filtering with delay compensation might be necessary to achieve satisfactory performance.

The fact that the MPC controller takes a certain amount of time to calculate the output command signal for the *k*-th step inevitably introduces delay in the signal propagation. The amount of delay depends on the control algorithm's computational complexity (e.g., length of the prediction horizon, total number of possible switching combinations, complexity of the IM model, etc.). There are other causes of delay, such as the D-A conversion, the IGBT driver circuit, the IGBT/diode pair itself (i.e., impossibility of instantaneous current commutation), the dead time, etc. Since all these delays typically sum up to a few microseconds, they could be easily compensated by utilizing the stator current prediction

for the $k + 2$ step instead of the $k + 1$ step for all possible switching states and by applying the optimal solution at the start of the next sampling period, provided that the sampling period is large enough.

The offset in the measured stator currents, which constitute periodic AC signals, could be detected and eliminated online by utilizing the fact that the corresponding mean value calculated for a certain integer number of periods must equal zero. Of course, this requires knowledge of the fundamental frequency and accurate zero-crossing detection.

The EMI noise represents a challenging problem where power converter switching and parasitic capacitances/inductances are involved. It causes signal distortion and may potentially lead to device malfunction, especially at higher switching frequencies. It can be suppressed by filtering (e.g., low-pass filters and/or common-mode chokes), shielding, and grounding.

The winding resistances are known to vary with both the temperature and frequency (i.e., skin effect). This variation could be assessed online by means of measurement (e.g., sensors installed in the windings) or through application of an observer (e.g., model-reference-adaptive system) and compensated through online correction of the resistance values in the control algorithm.

The MPCC sampling period utilized in this study falls within the typical range of a few tens of microseconds. Hence, it is expected that the proposed algorithm could be executed in real-time by using an affordable digital signal processor, especially considering the one-step prediction horizon, but this assumption is yet to be tested.

## 6. Conclusions

In this paper, several MPCC strategies based on IM models of different complexity levels have been presented and discussed. To our best knowledge, the MPC considered in this study is the only one that allows for inclusion of the IM magnetic saturation, iron losses, and SLLs, whereas it can be easily adapted to include only some of these phenomena. In addition, it is the only MPCC algorithm that includes *any* of these phenomena. It is also the first MPC algorithm based on the IM model in which the iron-loss resistance is placed in parallel to the *stator inductance*, instead of being placed in parallel to the magnetizing inductance. This simplifies the corresponding equations greatly, while not sacrificing much the accuracy.

Based on the extensive simulation analysis, the IM model including the magnetic saturation and rotor speed-dependent iron-loss resistance proved to be the best candidate for the MPCC design, providing the best tradeoff between the practicability and accuracy. The implementation of more complex IM models would imply a more complex process of parameter determination and increased computational costs, while not resulting in a substantially better performance. On the other hand, the implementation of simpler IM models would result in considerable performance degradation, the most important of which is the increase in the stator currents' THD (i.e., IM harmonic losses) and the average switching frequency (i.e., power converter losses). In addition, by penalizing the control effort in the proposed manner, the average switching frequency can be further reduced (i.e., more than twice with the utilized $\lambda_{sw}$ value), while not sacrificing too many other performance metrics. Compared to the IRFO control, the proposed MPCC with the CEP provides better overall dynamic performance, as well as lower torque ripples and stator current THD at rotor speeds above half the rated.

In the future, an experimental validation of the proposed method is to be carried out with special consideration for the CEP optimization and the effects of the measurement noise and signal delay. In addition, the impact of the IM winding resistances' variations is to be evaluated. A similar analysis is to be performed for varying IM magnetization levels (e.g., flux weakening or loss minimization) and for the MPTC-based IM drive.

**Supplementary Materials:** The following supporting information can be downloaded at: https://www.mdpi.com/article/10.3390/pr11102917/s1, These materials consist of the 3D surface plots of the considered performance metrics vs. the normalized load torque and rotor speed, obtained for the model variant *e*. In addition, the simulation parameters, the MPC code, and the simulation model segments in MATLAB-Simulink of the MPCC system based on the IM model variant *d* and including the CEP are provided for the purpose of reproducibility of the results. Figure S1: 3D surface plot of the stator current THD for the IM model variant *e*; Figure S2: 3D surface plot of the average switching frequency for the IM model variant *e*; Figure S3: 3D surface plot of the rotor flux magnitude error for the IM model variant *e*; Figure S4: 3D surface plot of the rotor flux angle error for the IM model variant *e*; Figure S5: 3D surface plot of the product $THD_I \cdot f_{sw\text{-}avg}$ for the IM model variant *e*; File S1: Simulation model parameters; File S2: MPC code in MATLAB; File S3: Simulation model segments in Simulink.

**Author Contributions:** Conceptualization, M.B.; methodology, M.B. and D.V.; software, M.B.; validation, M.B., D.V. and I.G.; formal analysis, M.B. and D.V.; investigation, M.B.; resources, M.B.; data curation, M.B., D.V. and I.G.; writing—original draft preparation, M.B.; writing—review and editing, D.V. and I.G; visualization, M.B. and I.G; supervision, D.V. and I.G.; project administration, M.B.; funding acquisition, M.B. All authors have read and agreed to the published version of the manuscript.

**Funding:** This research received no external funding.

**Data Availability Statement:** The data presented in this study are available on request from the corresponding author.

**Conflicts of Interest:** The authors declare no conflict of interest.

## Appendix A

The space-vector equations of the IM model shown in Figure 1a are obtained by substituting (6)–(8) into (1)–(5) and are given as follows ($\Sigma R = R_s + R_{sll} + R_m$):

$$\mathbf{v}_s = \mathbf{i}_s(R_s + R_{sll}) + \frac{d\boldsymbol{\psi}_s}{dt} + j\omega_e\boldsymbol{\psi} \tag{A1}$$

$$0 = R_r\mathbf{i}_r + \frac{d\boldsymbol{\psi}_r}{dt} + j(\omega_e - \omega_r)\boldsymbol{\psi}_r \tag{A2}$$

$$\boldsymbol{\psi}_s = \frac{L_s}{R_m}(\mathbf{i}_s\Sigma R - \mathbf{v}_s) + L_m\mathbf{i}_r \tag{A3}$$

$$\boldsymbol{\psi}_r = L_r\mathbf{i}_r + \frac{L_m}{R_m}(\mathbf{i}_s\Sigma R - \mathbf{v}_s) \tag{A4}$$

$$T_e = \frac{3}{2}p\frac{L_m}{L_r}\text{Im}\left\{\mathbf{i}_{sT}\overline{\boldsymbol{\psi}_r}\right\} = \frac{3}{2}p\frac{L_m}{L_r}\text{Im}\left\{\left(\frac{\mathbf{i}_s\Sigma R - \mathbf{v}_s}{R_m}\right)\overline{\boldsymbol{\psi}_r}\right\} \tag{A5}$$

## Appendix B

The polynomial equations describing the hysteresis loss coefficient $K_h$ (for accounted SLLs) and magnetizing inductance $L_m$ (in the saturated region) are given as follows:

$$K_h = -9.1403\frac{|\boldsymbol{\psi}_s|^2}{\psi_{s,rated}^2} - 10.6306\frac{|\boldsymbol{\psi}_s|}{\psi_{s,rated}} + 78.0902 \tag{A6}$$

$$L_m = 0.3457\frac{|\boldsymbol{\psi}_s|^3}{\psi_{s,rated}^3} - 1.4156\frac{|\boldsymbol{\psi}_s|^2}{\psi_{s,rated}^2} + 1.2905\frac{|\boldsymbol{\psi}_s|}{\psi_{s,rated}} + 0.0785 \tag{A7}$$

**Appendix C**

The current PI controllers' parameters were tuned according to the following procedure. First, the corresponding angular sampling frequency is chosen—in this case, $\omega = 2\pi \cdot 10$ kHz = 62,831.85 rad/s. According to the Nyquist stability criterion, the asymptotic stability is reached for the following closed-loop system bandwidth:

$$\alpha_c = \frac{\omega_s}{6} \tag{A8}$$

The recommended closed-loop system bandwidth is

$$\alpha_c < \frac{\omega_s}{10} \tag{A9}$$

which gives the phase and gain margins of 36° and 1.7, respectively (in this case, $\alpha_c < 6283.185$ rad/s).

For the first-order closed-loop system, by which the current control loops are approximated, the rise time is defined as

$$t_{rc} = \frac{\ln 9}{\alpha_c} \tag{A10}$$

where $\alpha_c$ equals the inverse value of the corresponding time constant.

From (A9) and (A10), it follows:

$$t_{rc} > \frac{10 \cdot \ln 9}{\omega_s} \tag{A11}$$

which yields $t_{rc} > 0.35$ ms in the considered case.

The desired $t_{rc}$ value in this study is set to 1.5 ms, which, in turn, yields $\alpha_c = 1464.82$ rad/s.

In the last step, by applying the direct synthesis method [47], the current PI controllers' gains are obtained as follows:

$$\begin{aligned} k_{pc} &= \alpha_c L \\ k_{ic} &= \alpha_c R \end{aligned} \tag{A12}$$

where $L = L_{sl} + L_{rl}$ and $R = R_{sT\text{-}rated} + R_r$

Note that $R_{sT\text{-}rated}$ in (A12) is obtained from (6) by setting $R_m = R_{m\text{-}rated}$ and $R_{sll} = 0$ (SLLs are neglected since only the IM model variant *d* is considered in Section 5.3).

The described procedure resulted in the following values of the current PI controllers' gains: $k_{pc} = 50$ and $k_{ic} = 11641$.

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
