# Peer review of "Model Predictive Current Control of an Induction Motor Considering Iron Core Losses and Saturation"

_processes, doi:10.3390/pr11102917_

Round 1

Reviewer 1 Report

The authors presented and discussed several MPCC strategies based on IM models of different complexity levels in this paper. 

Some minor corrections that will contribute to the quality of the article are listed below;

1. What function does the block with (14) & (15) written in it, quoted from reference 36 in Figure 5, fulfill? The expression specified in the block output does not match equations 14 and 15 given in reference 36. Here, it is recommended to add and use the equation given in reference 36 by explicitly referencing it in the text.

2. It is recommended to briefly interpret and explain the graphs in Figure 13. The graph in Figure 13c should be resized and given more clearly.

3. It is recommended that the results be interpreted by comparing them with the current literature.

4. It should be clearly stated what contribution the article makes to the literature on the subject in terms of its results.

5. The references used mainly consist of studies conducted in previous years. It is recommended to increase the number of current literature.

The manuscript is well-prepared and has a fluent turn of explanation. However, the article should generally be checked for spelling.

Author Response

We would like to thank the reviewer for his/her time and expertise in reviewing this manuscript. We found the comments to be valuable and important for the improvement of our paper. Our responses are given point-by-point in the attached document, whereas all the changes made are marked yellow in the revised manuscript.

Reviewer 2 Report

1) Provide the detailed mathematical equations that define the dynamic induction motor (IM) model, including magnetic saturation, iron losses, and stray-load losses (SLLs). Explicitly state the equations for key parameters such as iron-loss resistance and magnetizing inductance. 

2) While the paper discusses different levels of IM model complexity, provide a clear definition of what constitutes "complexity" in this context. What are the specific components or phenomena that contribute to the complexity of the model, and how are these incorporated?

3) Include more information about the simulation setup in Section 5. Describe the input conditions, boundary conditions, and any assumptions made during simulations. This will enhance the reproducibility of the results.

4) Discuss practical considerations and challenges in implementing the proposed model predictive controller in a real-world system. Consider addressing issues related to sensor noise, computational requirements, and any limitations that may arise in practical applications.

5) The introduction provides an overview of the paper but lacks clarity in explaining the main contributions and objectives. It is essential to explicitly state the paper's objectives, such as what specific problems or challenges the paper addresses and what outcomes are expected.

Author Response

(The authors gave the same response as above.)

Reviewer 3 Report

Dear Authors.

I revised the paper very carefully. I think that this is a very interesting and valuable paper with great scientific potential. This manuscript also has a practical value. The paper is written in a scientific, rigorous manner. The research is well-designed.

This is an original manuscript. I did not detect plagiarism or inappropriate self-citations.

The “Abstract” is short, descriptive, and concise. Also, the keywords reflects accurately the content of the paper.

In “Introduction” the critical literature review is presented. The advantages and disadvantages of the existing methods are briefly discussed. The model predictive control is quite a mature discipline, but the use of this method in the presented context is reasonable. Next, the mathematical model of the system is presented. Later, the proposed control system is described. In “Figure 5” you presented the detailed scheme of the control system. This is a very positive aspect of the research. Later, the Model Predictive Current Controller is proposed. Next, the obtained results are described. The performance of model predictive current control with various induction motor models was investigated. Such analysis is one of the most valuable portions of the presented research. The presented conclusions are supported by the obtained results. The performance of considered control algorithms was presented. The manuscript ends with a description of the findings. Also, the future possible research directions are suggested.

Additionally, the supplementary materials will be included with the final version of the paper. This is a very good idea. This is a key issue for achieving reproducibility of the results.

However, I think that the paper requires minor revision. In the proposed form, it cannot be accepted for publication. Below, please find my suggestions on how to improve the quality of the paper.

### General comments ###

*** The main contribution of the study is explained at the end of the paper. I suggest including a clear description of this issue in the “Introduction”. Of course, it might be concluded why the presented research is essential, but it will be much better if you include a brief explanation at the beginning of the paper.

*** The bibliography includes 44 references. All the cited publications are related to the topic. I think that you cite more papers. Some of them are “too old”. Maybe it will be better to cite papers from the last 5-7 years, but in the current form, the bibliography is also acceptable.

### Specific comments ###

*** Line 2

You mentioned that “…the dc machine, and typically…”. I suggest replacing “dc” by “DC” (using capital letters) or “direct current”.

*** It seems that in section two, “Induction Machine Modeling” the assumptions are not described sufficiently. Could you include a short paragraph with an explanation of this issue?

*** Figure 7, 8, 9, 10, 11

I suggest replacing the plots in Figures 7 to 11 with files in vector format or at least by Portable Network Graphics images in high resolution (at least 300 DPI). In the presented form, these pictures are rather too blurry.

*** Figure 13

Something is wrong with “Figure 13”. Could you replace the graphs with the appropriate ones? I think that this is only an editing mistake.

I highlighted (in yellow color) all my suggestions in the attached pdf file.

I wish you good luck with your study.

Kind regards,

Reviewer

English language is acceptable.

Author Response

We would like to thank the reviewer for his/her time and expertise in reviewing this manuscript. We found the comments to be valuable and important for the improvement of our paper. In addition, acknowledgment of our efforts and encouraging remarks are much appreciated. Our responses are given point-by-point in the attached document, whereas all the changes made are marked yellow in the revised manuscript.

Reviewer 4 Report

What I really liked in this paper is the model of the motor where Rm is moved over the stator leakage inductance. This helps a lot to reduce the complexity of the model without considerable impact to the accuracy. I didn't find the reference to such model in the literature. If this is unique invention of the authors, then this should be highlighted in the paper.

Authors didn't consider the continuous control set MPC like:

Varga, T.; Benšić, T.; Jerković Štil, V.; Barukčić, M. Continuous Control Set Predictive Current Control for Induction Machine. Appl. Sci. 2021, 11, 6230. https://doi.org/10.3390/app11136230

A. Anuchin, D. Aliamkin, M. Lashkevich, V. Podzorova, L. Aarniovuori and R. Kennel, "Model Predictive Control with Reduced Integration Step Size for Continuous Control of an IPM Motor," 2019 IEEE International Symposium on Predictive Control of Electrical Drives and Power Electronics (PRECEDE), Quanzhou, China, 2019, pp. 1-6, doi: 10.1109/PRECEDE.2019.8753231.

I suggest to make equations (11) and (12) more clear. The current derivative is dictated by the differential inductance for the voltage applied and impact of the back-EMF, which is the function of the speed and flux linkage normally defined by full inductance. Please clarify each symbol in the equations whether the full or differential inductance is used.

I have concerns about performance comparison of MPCC and IRFO. It is said that the sampling period for MPCC was 20 us, while speed control loop and update of the current reference was performed for both control strategies at 1 ms. This gives 50 times difference in the frequencies. Such difference is not good to judge the performance of the drive. Moreover, Figure 13b indicates that the current controller is badly tuned for IRFO. And this awful tuning impacts the performance of the speed loop for IRFO because overshoots in the speed transients are only cause by the current loop but not the speed loop performance.

Figure 13 is not readable. I can only guess what is depicted there. Its c) part is absent.

I suggest to accept the paper after minor revision.

Author Response

(The authors gave the same response as above.)

Reviewer 5 Report

Review of the article: “Model Predictive Current Control of an Induction Motor Considering Iron Core Losses and Saturation”

                The subject matter of this study and its findings could be of value to the Processes readership. The research conducted merits publication. Nevertheless, during the review process, I noted certain areas that require attention. Addressing these issues would enhance the paper's quality, and I also seek clarification on the following points:

1. The most important research results should be given in e.g. two sentences. This part of the data is missing in the Abstract.

2. It is recommended to improve the quality and readability of the figures, especially 13.

3. In the “Results and Discussion” section, there is no reference to literature sources in connection with the obtained research results.

The discussion of the results and the conclusions are very clear, therefore, the publication of the article on Processes Journal is recommended after the aforementioned small additions and modifications.

Author Response

(The authors gave the same response as above.)

Round 2

Reviewer 2 Report

Accept in present form.